# Defenses against Virus and Vector: A Phloem-Biological Perspective on RTM- and SLI1-Mediated Resistance to Potyviruses and Aphids

**DOI:** 10.3390/v12020129

**Published:** 2020-01-22

**Authors:** Karen J. Kloth, Richard Kormelink

**Affiliations:** 1Laboratory of Entomology, Wageningen University and Research, 6700 AA Wageningen, The Netherlands; 2Laboratory of Virology, Wageningen University and Research, 6700 AA Wageningen, The Netherlands; richard.kormelink@wur.nl

**Keywords:** potyviruses, aphids, plant resistance, phloem

## Abstract

Combining plant resistance against virus and vector presents an attractive approach to reduce virus transmission and virus proliferation in crops. *Restricted*
*Tobacco-etch virus Movement* (*RTM*) genes confer resistance to potyviruses by limiting their long-distance transport. Recently, a close homologue of one of the *RTM* genes, *SLI1*, has been discovered but this gene instead confers resistance to *Myzus persicae* aphids, a vector of potyviruses. The functional connection between resistance to potyviruses and aphids, raises the question whether plants have a basic defense system in the phloem against biotic intruders. This paper provides an overview on restricted potyvirus phloem transport and restricted aphid phloem feeding and their possible interplay, followed by a discussion on various ways in which viruses and aphids gain access to the phloem sap. From a phloem-biological perspective, hypotheses are proposed on the underlying mechanisms of RTM- and SLI1-mediated resistance, and their possible efficacy to defend against systemic viruses and phloem-feeding vectors.

## 1. Introduction

Transmission of vector-borne plant viruses in crops can be mitigated via several strategies, such as vector control or breeding for resistance to the vector or virus. In an optimal integrated pest and disease management plan, measures against both vector and virus are taken. This not only helps to minimize economic losses, but also to reduce the chance of resistance breaking. Breeding for plant resistance is often impeded by the lack of genetic markers or genetic resources of resistance, let alone the possibility of stacking several resistance genes against virus and vector in one cultivar. *Vat* is a rare example of a resistance gene that works against both *Aphis gossypii* and the viruses it transmits in melon, *Cucumis melo* [1,2]. Vat-mediated resistance has been extensively studied [3]. *Vat* genes encode nucleotide-binding leucine-rich repeat receptors that upregulate microRNAs and peroxidase activity in the mesophyll upon aphid infestation. Interestingly, recent studies have now revealed another group of plant genes that may have functional overlap in resistance to viruses and their aphid vector, but this time in the phloem. *Restricted Tobacco-etch virus Movement* (*RTM*) genes are one of the best characterized potyvirus resistance genes and limit long-distance transport in *Arabidopsis*. *SIEVE ELEMENT-LINING CHAPERONE1* (*SLI1*, located on chromosome 3) is a homologue of one of the *RTM* genes (*RTM2*, located on chromosome 5) and has recently been identified as a *Myzus persicae* aphid resistance gene in *Arabidopsis*. Both RTM- and SLI1-mediated resistance are phloem-located and involve transmembrane domain-containing small heat-shock-like (hsp20-like) proteins. Although both resistances have been found in the model plant *Arabidopsis*, homologues of *RTM* and *SLI1* seem to occur throughout several plant families [4,5,6]. The functional connection between restricted virus long-distance movement and restricted aphid phloem feeding raises the question of whether plants have a basic defense system in the phloem that acts upon generic to systemic virus spread and phloem-feeding insects. Still, very little is known about the nature of phloem-located resistance mechanisms. This review presents an overview of RTM-mediated restriction of potyvirus long-distance movement and SLI1-mediated resistance to aphids, followed by a discussion on hypotheses that would explain how phloem-located processes could affect potyviruses and their vectors.

## 2. Phloem Sap Transport and the Architecture of Sieve Tubes

While phloem tissue encompasses many specialized cell types, such as phloem parenchyma cells, companion cells, or sieve elements, this review focusses on sieve elements as the location of RTM and SLI1 proteins. Sieve elements are elongated cells that are interconnected into sieve tubes via callose-rich sieve plates [7] (Figure 1). Sieve tubes are the conduits for phloem sap. In past decades the old paradigm that sieve tubes function as pipelines for merely photo-assimilates has been replaced by the broader concept of conduits for sugars, ions, hormones, metabolites and signal molecules of various sizes, such as mRNAs, small (s)RNAs and proteins [8]. Apart from their conductive role in the transport of endogenous molecules, sieve tubes are also excellent gateways for viruses to disperse within a host and are productive food sources for insects, such as aphids and whiteflies. Phloem sap is allocated by a mass-flow mechanism which is generated by a hydrostatic pressure gradient that drives sap from source tissue, with high solute concentrations, to sink tissue, with low solute concentrations [9]. The location of source and sink tissues is flexible, as tissues change their sink or source status during development or infection incidence [10]. Unlike xylem vessels, sieve elements are alive and contain organelles, RNAs and a proteome. During differentiation, sieve elements enucleate and break down cytoskeletal elements, vacuoles, ribosomes and the Golgi apparatus [11,12]. What is left in a mature sieve element is a central lumen for intercellular sap transport, and the parietal layer, consisting of a thin cytoplasmic layer at the periphery of the sieve tube. Although mature sieve elements have lost their nuclear transcriptional apparatus and translational machinery, they contain vital organelles in the parietal layer, including endoplasmic reticulum, mitochondria and plastids [13]. The neighboring companion cells are important suppliers of proteins, RNAs and metabolites for sieve elements. Symplastic connections between sieve elements and their neighboring cells are limited in most plant species. Between sieve elements and phloem parenchyma cells, plasmodesmata are mostly absent. Between sieve elements and companion cells, the original plasmodesmata are shut off during cell differentiation [14] and later replaced by secondary plasmodesmata that, in *Arabidopsis* and many other plant species, branch on the companion cell side and have a single opening on the sieve element side [15]. These plasmodesmata at the companion cell–sieve element interface are also called the ‘plasmodesmata pore units’ [16]. Interestingly, plasmodesmata pore units can conduct molecules larger than 60 kDa and thereby are able to support the transport of macromolecules, such as proteins and long RNAs [17,18]. Conversely, not all molecules that are cell-to-cell mobile in epidermis and mesophyll can cross the plasmodesmata pore units, indicating that the plasmodesmata pore units are controlled by different movement factors in comparison to plasmodesmata in other tissues [11,19,20]. Presumably, this selective trafficking through plasmodesmata pore units may serve to prevent leakage of solutes from the phloem sap [15], regulate the long-distance communication of macromolecules within the plant [21,22], and protect the phloem sap from invading organisms [23].

## 3. Potyvirus Cell-to-Cell and Long-Distance Movement

Being the largest genus of plant viruses, potyviruses (family Potyviridae) afflict substantial economic losses in a wide range of crops [24]. Potyviruses are single-stranded, positive-sense RNA viruses that usually have a non-segmented genome encoding a polyprotein which is cleaved into 10 mature viral proteins. As they are able to infect many different tissues and cells, they are classified as systemic viruses. Most potyviruses are transmitted by aphids in a nonpersistent manner, i.e., aphids only need to probe for a few seconds in infected plants to acquire the virus at their stylet mouthparts [25]. Aphids mostly remain viruliferous for maximum several hours depending on feeding behavior. Virus transmission usually occurs during the first 15 seconds of a probe on a healthy new plant, when the aphid stylet is still in the epidermis or in the upper mesophyll tissue. After introduction into a new host plant potyviruses move from cell to cell in a stepwise process, where replication and protein expression is considered to take place before the virus proceeds to the next cell [26]. Cell-to-cell movement takes place via plasmodesmata [27], the cytoplasmic channels that interconnect plant cells and span the cell walls [28]. Currently, two models are commonly recognized to explain cell-to-cell movement of plant viruses. The first model involves movement of infectious virus particles or ribonucleoprotein (RNP) complexes in a tubule-guided fashion [27]. Tubules primarily consist of the viral movement protein (MP) and protrude from plasmodesmata of infected cells into the neighboring healthy cell. In a second model, viruses move along the cortical endoplasmic reticulum-actin network that is continuous between plant cells through plasmodesmata. Trafficking through plasmodesmata involves modification of the size exclusion limit and/or interaction with plasmodesmata-associated chaperones [21]. This process is thought to involve the MP which targets plasmodesmata for cell-to-cell transport of viral RNP or replication complexes [27]. Potyviruses do not encode a dedicated MP for size exclusion limit manipulation, nor construct their own tubule to travel through plasmodesmata [29]. Instead, they require the capsid protein (CP), helper component-protease (HC-Pro), cylindrical inclusion protein, and/or the P3N-PIPO protein for cell-to-cell movement [27,30]. Systemic viruses are considered to be unable to spread throughout the entire plant in a cell-to-cell mode due to tissue-specific barriers, such as the leaf abscission zone with high abundance of pathogenesis-related proteins [26]. For quick and effective systemic invasion, these viruses, including potyviruses, therefore rely on the phloem tissue. This involves a process distinct from cell-to-cell movement and is generally referred to as ‘long-distance transport’. When viruses reach the vascular bundle and enter a sieve tube, they are transported via the phloem sap over long distances to unload and subsequently infect systemic sink tissues. Although it is still unknown in which form potyviruses move through the phloem, long-distance transport of their RNA genome requires CP, HC-Pro and the viral protein genomic-linked (VPg) [30,31].

## 4. RTM-Mediated Resistance to Potyviruses

The *RTM* gene complex in *Arabidopsis* has been the focus of research for more than 20 years and prevents long-distance transport of potyviruses. The *RTM* genes convey dominant resistance to tobacco-etch potyvirus (TEV) [5,32], lettuce mosaic virus (LMV) and plum pox virus (PPV) [33], and may be associated with quantitative resistance to turnip mosaic virus (TuMV) [34]. Potato potyvirus Y (PVY), tobacco vein mottling virus (TVMV) and cucumber mosaic cucumovirus (CMV) do not seem to be affected by RTM-mediated resistance [5]. The *RTM* gene complex consists of at least five genes, of which three (*RTM1*, *-2*, and *-3*) have initially been mapped in *Arabidopsis* exhibiting differences in systemic spread of TEV [32]. Experiments showed that after mechanical inoculation of the upper cell layers of leaves with TEV, each of these three genes are required for restriction of virus long-distance transport to the inflorescence and distal leaves. The RTM-mediated resistance is ‘atypical’, while there is no hypersensitive response, no induction of *PATHOGENESIS-RELATED GENE 1* (*PR1)*, and unaffected local cell-to-cell movement. Only long-distance movement, from the local site of infection to the inflorescence, is compromised. Follow-up studies resulted in the cloning of *RTM1*, *-2* and *-3* [5,6,35,36,37] and the identification of the *RTM4* and *RTM5* QTL loci [38]. *RTM1* to *RTM3* encode various types of proteins from rather obscure families: a jacalin-repeat protein (RTM1), a transmembrane-containing small heat-shock-like (hsp20-like) protein (RTM2), and a meprin- and tumor necrosis factor receptor–associated factor (TRAF)-domain containing protein (RTM3). Interestingly, recent insights reveal that jacalin-like lectins, such as RTM1, are often involved in plant defense and may function as a decoy to trap pathogens [39]. Hsp20-like proteins, such as RTM2, have diverse tissue-specific functions and are often involved in protein folding, preventing protein aggregation and in degrading aberrant proteins [40]. The family of meprin- and TRAF-domain containing proteins, to which RTM3 belongs, was previously undescribed and is related to proteins with receptor-binding properties involved in necrosis [36,41,42]. RTM1 and RTM2 proteins localize to sieve tubes where they occupy the parietal layer, which is the small cytoplastic layer along the sieve tube plasma membrane. RTM1 tends to form distinct protein bodies with a diameter of 1 µm, while RTM2 lines the sieve tube wall in either punctuate or continuous pattern and sometimes forms circular shapes [35,43]. The localization of RTM3, which expresses at lower levels relative to RTM1 and -2, is as yet unknown. Interestingly, Decroocq et al [44] observed that changes in the N-terminus of the PPV and LMV CP associated with breaking of the RTM-mediated resistance. This led to the hypothesis that the *RTM* gene complex prevents long-distance transport via the sequestration or obstruction (of certain domains) of the viral CP. So far, however, no interactions have been observed between RTM proteins and the CP or the VPg of LMV [45]. Yeast-two hybrid assays only have revealed monomeric and multimeric interactions of RTM1 and RTM3, but not of RTM2 [36]. Therefore, a larger multimeric protein complex has been postulated, containing additional yet unidentified host proteins, which would (indirectly) link the RTM proteins to viral CP. During a recent large yeast-two hybrid screen of a companion cell cDNA expression library [45] LATERAL ROOT PRIMORDIUM 1 (LRP1), BRANCHED-CHAIN AMINOTRANSFERASE4 (BCAT4) and FIBRILLIN 1B (FBN1b) have been identified to interact with both RTM proteins and LMV CP. These proteins are involved in plant development, aliphatic glucosinolate biosynthesis and plastid lipids, respectively. However, functional analysis of T-DNA insertion mutants of *Arabidopsis* showed that none seem to be uniquely required for RTM-mediated resistance.

## 5. SLI1-Mediated Resistance to *Myzus persicae* Aphids

Recently, *SLI1*, a close homologue of *RTM2*, has been characterized as a resistance gene for the green peach aphid, *Myzus persicae*, a vector of TEV [46]. *SLI1* was identified via genome-wide association mapping of aphid feeding behavior on 350 natural *Arabidopsis* accessions. Associations between the duration of aphid probes and mutations on chromosome 3 of *Arabidopsis* revealed a thus-far-unknown resistance gene that lead to reduced aphid reproduction. *SLI1* encodes, similar to *RTM2*, a hsp20-like protein with a predicted transmembrane domain and localizes only in sieve tubes, where it occupies the parietal layer of the sieve tube and sieve plates. In protoplasts, SLI1 co-localizes with the endoplasmic reticulum. Unlike *RTM2*, *SLI1* shows a heat-responsive expression, and the effects on aphids are severest under moderate heat stress of 26 °C. Although SLI1 does not confer absolute resistance, it reduces aphid reproduction by almost 50% at 26 °C. Detailed electrical penetration graph recording of aphid feeding behavior has shown that aphids spent less time phloem-feeding and more time salivating in the sieve tube on plants containing SLI1. When feeding, honeydew excretion rates are lower, indicating that SLI1 reduces the rate of phloem sap ingestion. Upon extended salivation into the phloem, aphids are able to overcome natural variation in SLI1-mediated resistance between *Arabidopsis* accessions [46]. Based on these observations, SLI1 was hypothesized to be involved in plugging the entrance of the aphid food canal with protein and/or membrane agglomerations. Salivation overcomes SLI1-mediated resistance, most likely via aphid saliva containing effectors, such as calcium-binding proteins and proteinases that could inhibit or degrade these obstructions [47].

## 6. What Are the Underlying Mechanisms of RTM- and SLI1-Mediated Resistance?

Even though the targets of RTM- and SLI1-mediated resistance are completely different (potyviruses, respectively aphids) and their resistance leads to a different outcome (restriction of virus long-distance movement and restriction of aphid phloem sap ingestion, respectively), there are several commonalities to be found. Both *RTM2* and *SLI1* encode a transmembrane domain-containing hsp20-like protein. The site of action of both gene products is in the sieve tubes, and their mode of action is to block phloem sap accession. These similarities raise the intriguing question on whether RTM- and SLI1-mediated resistances share a common physiological machinery against biotic intruders. The following sections will propose mechanisms that could affect both potyviruses and aphids, in the order of four major physiological barriers in the sieve tube.

### 6.1. The Companion Cell–Sieve Element Interface (Loading)

The companion cell–sieve element interface is the first barrier to be addressed. RTM1, RTM2 and SLI1 proteins localize in the parietal layer of the sieve tube, which is a region closely associated to companion cells via plasmodesmata pore units and channels for apoplastic transport between companion cells and sieve elements. Below, the possible implications for virus loading and the import of defense molecules are discussed.

#### 6.1.1. Restricted Viral Entry into the Sieve Tube

A prerequisite for long-distance movement of plant viruses via the phloem sap is that viruses enter the sieve tube. Many studies underline that virus movement into, and along sieve tubes, is different from local cell-to-cell movement. In the mesophyll, cell-to-cell spread takes place within hours, while virus movement towards the phloem sap can take days. One of the reasons for the latter might relate to the absence of a translational machinery in the endomembrane system of mature sieve elements. Although some evidence for viral replication in sieve tubes has been reported [48] (further discussed in Section 6.2.1), the lack of a translational machinery in sieve tubes may require the transport of viral proteins through plasmodesmata pore units. Translation of viral systemic movement factors, such as CP, would in this scenario occur in companion cells and these would associated to, or separate from viral RNP complexes, require transport into the sieve tube [49]. Another reason involves specific properties of plasmodesmata pore units that may require additional (viral) movement factors different from those needed to pass plasmodesmata, or the requirement of plasmodesmata pore unit-specific chaperones [23,50,51]. RTM proteins may affect the movement of potyviruses across the companion cell–sieve element interface (Figure 2). As shown by Chisholm et al. [35] and Cayla et al. [43], RTM1 and RTM2 proteins are confined to the parietal layer of sieve elements. Subcellular localization was, however, not resolved and co-localization with the plasmodesmata pore units at the companion cell–sieve element interface remains unknown. Neither has been investigated to determine whether RTM proteins are able to cross the plasmodesmata pore units by themselves. The occasional detection of RTM1 protein bodies in cells directly adjacent to sieve element [35], which are presumably companion cells, suggest that either RTM1 is expressed in both companion cell and sieve element, or, that it is able to modulate the size exclusion limit of plasmodesmata and cross the companion cell-sieve element interface. If RTM2 is localized in the endoplasmic reticulum similar to its homologue SLI1 [46] it would have an (in)direct connection to endoplasmic reticulum-derived desmotubules in the plasmodesmata pore units. The RTM complex could then restrict virus movement through the plasmodesmata pore units by, for example, competing for the binding site with a virus movement factor, resulting in (partial) sequestering of viral entities/RNPs (in the plasmodesmata pore units), or by blocking the interaction with endogenous chaperones required for movement through the plasmodesmata pore units.

#### 6.1.2. Influx of Ions and Loading of Defense Molecules

For aphids, the sieve element-companion cell boundary might be relevant from a different perspective. Using electrical penetration graph recordings it has been shown that SLI1-mediated resistance does not affect the success or duration for aphids to reach a sieve tube and penetrate the sieve element plasma membrane. Problems occur only thereafter, as illustrated by extended salivation in the sieve element and prematurely interrupted phloem feeding sessions [46]. This could be caused by ions, metabolites, RNAs and proteins that originate from the surrounding of the sieve tube and are imported via, e.g., transporters, channels and plasmodesmata pore units (Figure 2). Ca^2+^ influxes in sieve tubes, for example, are gated through ion channels, such as the GLUTAMATE RECEPTOR-LIKE3.3 (GLRs) [52,53]. It is known that wound- or temperature-induced Ca^2+^ influxes can decrease sieve tube conductivity in Fabaceae [54] and that aphids have been shown to counteract these influxes by the secretion of calcium-binding protein-containing watery saliva [47,55]. The import of detrimental secondary metabolites into the sieve tube may also be involved, such as phloem-transported camalexins [56] and glucosinolates [57] which both function as antifeedants for aphids. Glucosinolates are stored in cells surrounding the sieve tube, imported in companion cells via GLUCOSINOLATE TRANSPORTERS (GTRs), and subsequently loaded into the sieve tube [58,59]. Other potential defense factors, e.g., small interfering RNAs (siRNAs) and proteases, are considered to be loaded in the sieve tube via the plasmodesmata pore units [17,21]. SLI1 is localized at the sieve element endoplasmic reticulum, which is in contact with plasmodesmata pore unit desmotubules and covers almost the complete sieve element plasma membrane [60], and therefore might be in close contact with ion channels. A facilitating role of SLI1 in the loading of detrimental secondary metabolites, siRNAs or resistance proteins via the plasmodesmata pore units or its involvement in regulating Ca^2+^ influxes could explain the SLI-mediated increase in aphid salivation and reduction in feeding behavior [46]. Such a functionality would (indirectly and simultaneously) affect virus loading into sieve elements as well. In relation to this, it is interesting to note that camalexin has earlier been found to be associated to a restriction in systemic spread of cauliflower mosaic virus (CaMV) [61].

### 6.2. The Parietal Layer–Phloem Sap Interface

Rarely discussed is that entrance into sieve tubes does not necessarily lead to contact with phloem sap. The sieve element reticulum, which consists of ribosome-devoid endoplasmic reticulum, coats nearly the complete plasma membrane in the sieve tube with stacked cisternae and single membrane sheets. There is, thus, seemingly no direct contact between the sieve element plasma membrane and the phloem sap, except within scattered circular openings of approximately 80 nm in diameter in single sieve element reticulum membrane sheets [60]. This implies that viruses which have entered the sieve tube via plasmodesmata pore units, or aphid stylets that have punctured the sieve element plasma membrane, do not necessarily enter the phloem sap directly, but may first arrive in the inner layers of the sieve element reticulum (Figure 3). This ‘sieve element reticulum microenvironment’ between plasma membrane and lumen is suggested to serve plants as an important supplier of cations and ATP in exchange for apoplastic loading of sucrose and as a protective interface to prevent molecules from being translocated by the phloem sap flow [15,16,60].

#### 6.2.1. Sieve Element Reticulum Defenses against Virus Replication and Assembly

Following the principles of cell-to-cell movement, viruses may need to accumulate to critical levels and assemble before actual long-distance movement can take place [23]. Once in the sieve element, viruses may use the sieve element reticulum, phloem mitochondria or phloem plastids to this end. Evidence for this has been obtained from beet western yellows virus (*Luteoviridae*) [62], cucumber mosaic virus (*Bromoviridae*) [49], and turnip mosaic virus (*Potyviridae*) [48]. From these viruses, particles and CP have been found in high densities in the sieve element reticulum and parietal layer, some in membrane-enveloped structures. In light of these observations Blackman et al. [49] hypothesized a three-step process: (1) MP-guided trafficking through the plasmodesmata pore units, (2) virus assembly in a dedicated sieve element reticulum microenvironment, and (3) rupture of the membrane and release of virions into the phloem sap. For turnip mosaic virus (TuMV) aggregates of the small potyvirus 6K_2_ protein, which is able to use endoplasmic reticulum-derived membranes and to induce vesicle formation [63,64], have been observed in sieve tubes [48]. These 6K_2_ bodies, which co-localize with lipids, viral RNA-dependent RNA polymerase (RdRp) and double-stranded (ds)RNA, indicate that TuMV replication takes place in sieve element reticulum membrane-derived structures prior to long-distance transport [48]. Although reported, it remains questionable how replication can take place in absence of native ribosomes in the sieve element reticulum [11,12]. Nonetheless, potyvirus replication and the accumulation of viral dsRNAs may induce a (local) RNA silencing response in the sieve tube. Although several studies show no or negligible RNase activity in phloem sap exudates [65,66], emerging evidence indicates that animal and plant RNA silencing might be tightly linked to elements of the endomembrane system [67], which is present in sieve tubes in the form of sieve element reticulum. Potyviruses are prone to antiviral RNAi that partly relies on RNA-DEPENDENT RNA POLYMERASE 1 (RDR1) and RNA-DEPENDENT RNA POLYMERASE 6 (RDR6) amplification, in which the viral HC-Pro protein acts as a silencing suppressor and enables successful long-distance transport and systemic infection [68]. As RTM1 and RTM2 localize in the parietal layer of the sieve tube [35,43,46], they could be involved or interfere in the above described processes in various ways. A first possibility could involve inhibition of 6K_2_-mediated membrane proliferation and vesicle formation by RTM, thereby preventing the establishment of viral replication and/or assembly complexes. Secondly, RTM proteins may be required for sieve-tube located RNA silencing, to repress potyvirus replication. RTM proteins could, for example, be required for recruitment of dicer-like proteins or components of the RNA-induced silencing complex to the viral replication bodies. Since the N-terminus of the CP has been shown to play a key role in RTM resistance [44], a third possibility could involve an interference, by RTM, on the assembly or trafficking of a virus movement complex. However, so far, evidence is lacking that points towards the formation of a multimeric protein complex in which viral CP and RTM proteins (indirectly) interact [45], and which would support this third possible idea.

#### 6.2.2. Parietal Occlusion Mechanisms against Aphids

For *M. persicae* aphids, it has been suggested that SLI1 is involved in occluding the food canal with sieve element reticulum membranes and/or proteins [46]. Aphids are passive phloem-feeders [69]. After puncturing the sieve element plasma membrane, they secrete watery saliva before turgor-driven phloem sap enters the food canal. Salivation in the sieve tube is considered to counteract wound-induced responses or to chelate Ca^2+^ ions that induce protein aggregations [47,70]. However, when salivation is not effective, sieve element reticulum membranes or wound-induced protein aggregations could be pushed against the aphid stylet tip by the turgor pressure of the sap [46]. In this scenario, SLI1 could be involved in rapid Ca^2+^ accumulation in the sieve element reticulum or Ca^2+^ release from the sieve element reticulum to induce protein coagulation. Or, SLI1 could function as a scaffold for multimeric protein complexes that clog the aphid food canal. Alternatively, it could be involved in the sequestration and/or proteolysis of aphid salivary effectors that otherwise would degrade proteinaceous or membranous obstructions.

#### 6.2.3. Plastid- and Mitochondria-Induced Defenses Against Viruses and Aphids

Attachment to the chloroplast membrane is a well-known infection signature of many viruses, and is thought to play a role in counteracting chloroplast-induced defenses and viral replication [71]. As plastids and mitochondria are the only sites of transcription in mature sieve elements, they could play a central role in defense responses and suppression by viruses. Little is known about the physiological role of sieve-tube located plastids and mitochondria. Phloem plastids contain starch and/or proteins [72]. Upon sudden pressure changes in the sieve tube, plastids can burst and release their content into the phloem sap and obstruct sieve plate pores [73]. Sieve-tube located mitochondria are often embedded in the sieve element reticulum and can be expected to have a similar functionality as other mitochondria; e.g., to supply ATP and Ca^2+^, to have intensive exchange of ions and molecules with the sieve element reticulum via contact sites, and to import and degrade (misfolded) proteins from elsewhere in the cell [74,75]. Although the organellar associations of RTM1 and RTM3 in sieve tubes are unknown, both proteins localize at the perimeter of chloroplasts when transiently expressed in tobacco epidermal cells [45]. In sieve tubes, RTM1 forms spherical bodies with a diameter comparable to phloem plastids [43]. Furthermore, upon transient co-expression of RTM2 and BCAT4, a binding factor of RTM1 and LMV CP, in tobacco epidermal cells RTM2 becomes recruited to the chloroplast perimeter [45]. A role of the RTM complex in plastid-induced local defenses and/or interference with viral suppression of these defences can, therefore, not be ruled out yet. For SLI1, localization at the periphery of sieve tube mitochondria has been observed, but this was not the case for plastids [46]. Although speculative, SLI1-mediated resistance to aphids could involve transcriptional or (post-)translation processes in sieve tube mitochondria. As SLI1 is a hsp20-like chaperone, which are renowned for their involvement in proteolysis [40], degradation of aphid effectors or endogenous suppressors in mitochondria or the sieve element reticulum could be one of the possibilities.

### 6.3. Limiting Factors in the Phloem Sap

Once viruses or aphids have gained access to the phloem sap, they can travel or feed, respectively, passively due to the pressure-driven sap flow. They do not need to fear threats such as RNase or proteolysis activity, as these processes seem absent or negligible in the phloem sap [65,66,76,77]. There are, however, two other resistance factors to take into account during the phloem sap stage.

#### 6.3.1. Phloem-Mobile Virus Chaperones

One of the unresolved issues of phloem mobility is whether viral molecules require certain motifs or chaperones to hitchhike and move along with the phloem sap flow. The broad range of phloem-mobile mRNAs suggests that there may not be any required phloem-specific motifs for long-distance transport of host nucleotides [78]. There is, however, evidence that viruses are chaperoned by phloem-mobile host proteins [79,80,81]. If so, and potyviruses travel as virions that require chaperones for their movement in sieve tubes (Figure 4), interference to these interactions by RTM proteins could affect long distance transport.

#### 6.3.2. Sieve Tube Occlusion

An important factor for the success of both virus transport and aphid phloem sap ingestion is sieve tube conductivity, i.e., the ease with which sap volumes can move through the tubes. Plants can adjust the sieve tube conductivity via sieve plates. Pores on the sieve plates are ontogenetically modified plasmodesmata [82] and are much wider than ordinary plasmodesmata, but their number and width depend on their developmental stage and location in the plant. In *Arabidopsis*, sieve pores have a diameter of approximately 1 µm, but in some families, such as *Cucurbitaceae*, they can even reach up to 5 or 10 µm [83]. The pore size is highly plastic and thereby able to modulate the conductivity of sieve tubes [84]. Upon wounding or other stress factors, plants adjust the pore size within minutes by deposition of callose collars and thereby reduce or virtually shut off sap transport in a specific sieve tube [82]. This process is reversible, as callose plugs on the sieve plate can be broken down in several hours [85]. Also, fibrillar and filamentous phloem protein structures that disperse upon pressure perturbations can occlude the sieve pores. There are many examples where wounding or temperature shocks resulted in immediate dispersal of these proteins to the lumen, where they were pushed against the sieve plate [13,85,86,87,88]. Both callose deposition and phloem protein dispersal are considered to contribute to restricted virus transport, reduced sap availability for aphids, and occlusion of the aphid food canal [47,89,90]. A role of RTM2 and SLI1 in callose deposition is not unlikely (Figure 4). Both RTM2 and SLI1 localize on sieve plates [35,46]. SLI1 is most likely associated with the sieve element reticulum, which resides at the sieve pore margins and is thought to be involved in callose deposition [12,91]. Although no effects have been observed of SLI1 on callose accumulation in leaf veins with aniline blue staining [46], these observations still have to be verified with refined ultrastructural observations of sieve plates in healthy and stressed plants. Since RTM1, RTM2 and SLI1 are members of protein families (jacalin-like and hsp20-like, respectively), which are renowned for their ability to form multimeric protein complexes [92,93] they could as well be involved in the proteinaceous obstruction of sieve pores.

### 6.4. The Sieve Element–Companion Cell Interface (Unloading)

The last barrier to address in terms of RTM- and SLI1-mediated resistance is the site where unloading from sieve tube to companion cell takes place. This is relevant for both virus systemic spread, as well as the unloading of defense compounds in distal tissues for systemic defenses against aphids and viruses (Figure 4).

#### 6.4.1. Restricted Virus Unloading

Getting in or out a sieve tube seems to be two sides of the same coin. Interestingly, viruses load into the phloem of many kinds of veins, but unload only from major veins, suggesting that there are different factors involved in loading and unloading [31,94]. This is supported by studies that have reported the requirement of TMV-interacting pectin methylesterase to exit but not enter the phloem [95], and the requirement of turnip crinkle virus (TCV) (*Tombusviridae*) virion assembly for unloading [96]. The latter suggests a facilitating role of CP to exit the phloem. As the CP has also been shown to be a key factor for breaking RTM-mediated resistance for LMV and PPV [44], it would be very informative to find out whether RTM resistance restricts loading and/or unloading of potyviruses. In case of unloading, RTM proteins could be involved in similar processes as described above, considering that the same barriers need to be crossed in a reverse manner: the parietal layer and the plasmodesmata pore units. There is, however, another exit. If viruses follow a sieve tube to the end, they end up in immature sieve elements which are still in the process of differentiation [23,97]. Here, the subcellular organization is completely different, and might still include an extended cytoplasmic structure and a nucleus. SLI1 proteins have been observed to localize in these cells throughout the symplast and seem to be present at high titers (unpublished data, Kloth et al.). If this is the case for RTM proteins as well, their ubiquitous abundance might increase the chance of getting into contact with viral entities, and enhance their chance of sequestering the virus and preventing its spread into systemic tissue.

#### 6.4.2. Unloading of Systemic Resistance Factors

In contrast to the scenarios above, which depict a localized resistance mechanism, RTM genes have been proposed to induce systemic defenses [6,32]. In this case, signals or defense molecules could travel ahead of the infection via the phloem sap, unload in distal tissue, and prevent successful systemic establishment of potyviruses. The nature of such an RTM-mediated resistance is unknown and does not involve hypersensitive response or systemic acquired resistance pathways via *PR1* [32]. Alternatives for RTM-mediated systemic resistance could, for example, involve the unloading of defense metabolites such as camalexin, siRNAs that would prime RNA-induced silencing complexes, or signals that trigger accumulation of callose deposition or reactive oxygen species in distal tissue. For aphids, systemic effects of SLI1 have not yet been investigated. Although assays on aphid feeding behavior show local effects [46], induced effects at the whole-plant level cannot be excluded. Unloading of any signals or molecules from the phloem that would trigger the accumulation of callose, reactive oxygen species or toxic metabolites such as camalexin and indole glucosinolates, would reduce aphid performance as well. Before studying any of these processes, first it needs to be clarified if RTM- and SLI1-mediated resistance act systemically at all.

## 7. Conclusions and Future Outlooks

So far, clear evidence is lacking to point to any of the above described barriers being involved in RTM- and SLI1-mediated resistance toward viruses and aphids, respectively. Until then, none of these barriers can be ruled out for the simple reason that the present genuine physiological hurdles to be taken by aphids, in order to feed on phloem sap, and viruses, to systemically infect their host plant. In light of *RTM2* and *SLI1* being homologous, the intimate relation between aphids and viruses in the process of gaining access to the phloem sap, it is not very surprising to see some overlap in phloem-based virus and aphid resistance. But are we facing two sides of the same coin? In other words, are we dealing with one and the same resistance mechanism? This will be an intriguing question to solve. Still, many details of sieve-tube located processes in relation to long-distance virus movement and aphid feeding remain elusive. While for potyviruses it is known that the viral proteins CP, VPg and HC-Pro play a role in long-distance transport via sieve-tubes, their exact roles still need to be determined. Additionally, it is not known in which form they enter sieve tubes, if they replicate and assemble in sieve tubes, how they transport through the sap, and how they unload. Even less is known on SLI1-mediated resistance to *M. persicae* aphids, as this field of research has just emerged. For now, the only observation that currently directs our thoughts on a possible underlying resistance mechanism relates to RTM2 and SLI1 being homologous and phloem-located. For these reasons, it is tempting to speculate on the possibility of both RTM- and SLI1-mediated resistance being based on the same underlying mechanism and affecting both potyviruses and aphids. To elucidate the underlying mechanism(s), more data is needed on in-situ localization of both resistance proteins to more precisely identify the site of action. Proteomic analyses need to be continued to identify interacting host proteins that will help to pinpoint to (multimeric) protein complexes and possible processes involved in healthy and infected sieve elements. It will be a major challenge, considering that the phloem is not easily accessible, contains low gene expression levels and a high vulnerability for artefacts [73]. The next challenge will be to find out whether these resistance mechanisms would also affect long-distance transport of other viruses and their corresponding phloem-feeding insect vectors, including systemic- and phloem-limited viruses (e.g., potyvirus versus luteoviruses), but also DNA viruses, and other aphid and whitefly species. Such findings would turn phloem-based resistance into a very powerful source to combat a virus, its insect vector and virus transmission in one go.

## Figures and Tables

**Figure 1 viruses-12-00129-f001:**
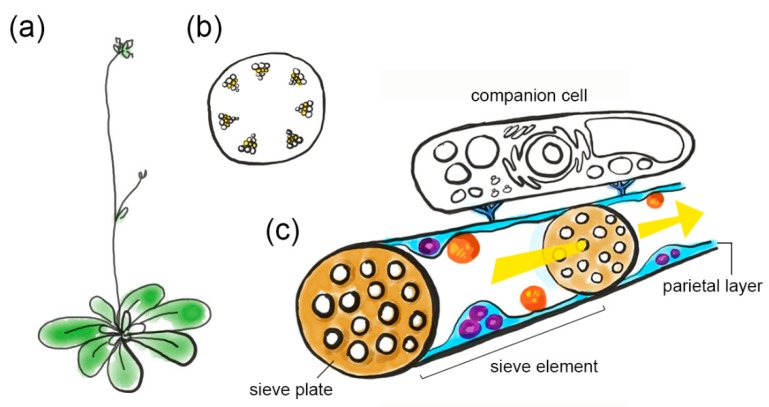
An impression of phloem architecture. (**a**) *Arabidopsis* plant. (**b**) Transverse section through the inflorescence stem, showing vascular bundles (yellow is sieve tube region). (**c**) Sieve tube with phloem sap allocation (yellow arrow) through sieve elements, separated by sieve plates. Sieve elements contain a parietal layer with phloem plastids (orange) and mitochondria (purple) that are embedded in the sieve element reticulum (light blue). Sieve element plasma membrane and cytoplasm are not depicted separately. Sieve elements are connected to a companion cell (colorless) via plasmodesmata pore units (dark blue).

**Figure 2 viruses-12-00129-f002:**
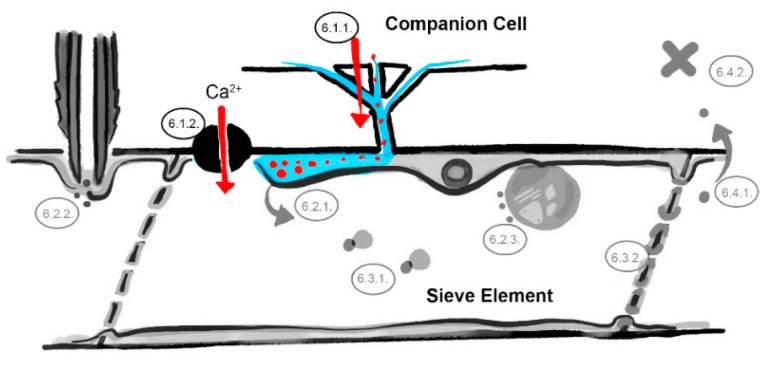
The companion cell–sieve element interface barrier. Restricted transport of viruses through plasmodesmata pore units (Section 6.1.1) and influx of ions and loading of defense molecules (Section 6.1.2). Involved barriers are indicated in color (light blue = sieve element reticulum, red dots = virus identities, black = ion channel, numbers correspond to the listed mechanisms in the main text, image not drawn to scale).

**Figure 3 viruses-12-00129-f003:**
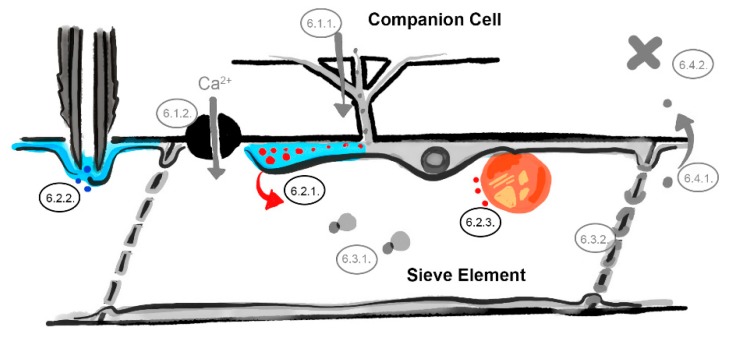
The parietal layer–phloem sap interface barrier. Defenses in the sieve element reticulum (Section 6.2.1), occlusion of aphid stylets (Section 6.2.2), and plastid-located defenses (Section 6.2.3). Involved barriers are indicated in color (light blue = sieve element reticulum, dark gray = aphid stylets, dark blue = aphid saliva, red dots = virus identities, orange sphere = phloem plastid, numbers correspond to the listed mechanisms in the main text, image not drawn to scale).

**Figure 4 viruses-12-00129-f004:**
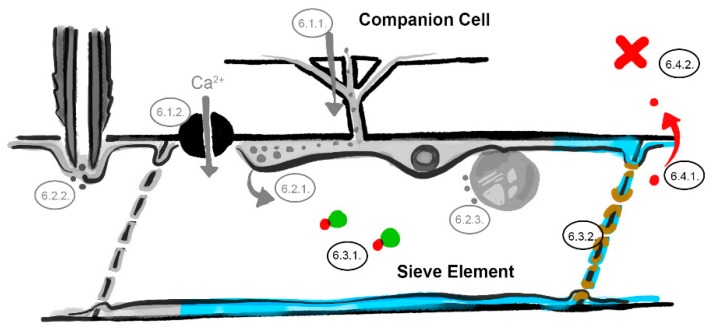
The phloem sap and sieve element–companion cell interface barriers. Interference with phloem-mobile chaperones (Section 6.3.1), occlusion of the sieve plate pores (Section 6.3.2), restricted virus unloading (Section 6.4.1) and systemic resistance factors (Section 6.4.2). Involved barriers are indicated in color (light blue = sieve element reticulum, red = virus identities, green = chaperone, brown = callose; numbers correspond to the listed mechanisms in the main text, image not drawn to scale).

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
