# Peer review of "Defenses against Virus and Vector: A Phloem-Biological Perspective on RTM- and SLI1-Mediated Resistance to Potyviruses and Aphids"

_viruses, 2020, doi:10.3390/v12020129_

Round 1

Reviewer 1 Report

The manuscript entitled "Defenses against virus and vector:a phloem biological perspective on RTM and SLI1-mediated resistance to Potyviruses and aphids" has discussed the Potyvirus phloem transportation and aphids phloem feeding. 

The manuscript is written in the view of  phloem biological, and proposed a model to explain the possible mechanisms of RTM- and SLI1-mediated resistance. The model sounds quite interesting. The manuscript is well written except the references part. Please pay more attention to the reference list and revised it one by one.

Author Response

Thank you for your comments. We have revised the reference list.

Reviewer 2 Report

Dear authors,

I found this review very interesting and I learnt a lot. I only have few suggestions.

In the figures, as in text, I do not like to have too many abbreviations what hinders a fluid lecture. I would prefer to read complete words for all structural elements (sieve element, reticulum …) and read abbreviations only for genes, proteins, viruses...

I did not understand at all the sentence in the lines 117-118 and the sentence in the line 452

I was wondering what is the localization on the Arabidopsis genome for SL2 and RTM2

Several times you wrote that potyviruses may replicate in sieve-tube (Lines 229 or 442) but I guess  that absence of a translational machinery in sieve tube obliterates virus replication in sieve tube.

There are numerous hypothesis presented in the paragraph 6 but I did not find the figure two very informative from that point. I would suggest designing one figure for each $, 6.1, 6.2…. with a common framework and try to design the most relevant hypothesis for RTM and SLI actions at each stage.

Please check you use again all abbreviations you wrote like ribonucleoprotein (RNP)

Sincerely

Author Response

Thank you for your comments and suggestions.

We have removed the abbreviations for organelles and structural elements. Apart from gene names and virus names we only kept abbreviations for commonly used terminology in virology, such as siRNAs, RNPs, MP. With this, we think the readability has improved.

Following your question, we have re-written the sentences in lines 120-122 and 497.

The genomic location of SLI1 and RTM2 is now added in lines 40 and 41.

You are correct about the unclarified issue of absence of translational machinery and viral replication. From ultrastructural studies it is apparent that there are no ribosomes in the sieve element reticulum of mature sieve elements. There is however evidence for virus replication in the sieve element reticulum from at least one study. We address this unclarified issue now in lines 255-258 and line 319.

We have split up Figure 2 into three separate figures to make it more comprehensive, and we suggest the editors to insert the images above the paragraph where the involved barrier is discussed.